# Fair, Polylog-Approximate Low-Cost Hierarchical Clustering

**Marina Knittel**   **Max Springer**   **John Dickerson**   **MohammadTaghi Hajiaghayi**
Department of Computer Science
University of Maryland, College Park
`{mknittel,mss423}@umd.edu`
`{john,hajiagha}@cs.umd.edu`

## Abstract

Research in fair machine learning, and particularly clustering, has been crucial in recent years given the many ethical controversies that modern intelligent systems have posed. Ahmadian et al. [2020] established the study of fairness in *hierarchical* clustering, a stronger, more structured variant of its well-known flat counterpart, though their proposed algorithm that optimizes for Dasgupta's [2016] famous cost function was highly theoretical. Knittel et al. [2023] then proposed the first practical fair approximation for cost, however they were unable to break the polynomial-approximate barrier they posed as a hurdle of interest. We break this barrier, proposing the first truly polylogarithmic-approximate low-cost fair hierarchical clustering, thus greatly bridging the gap between the best fair and vanilla hierarchical clustering approximations.

## 1   Introduction

Clustering is a pervasive machine learning technique which has filled a vital niche in every day computer systems. Extending upon this, a *hierarchical clustering* is a recursively defined clustering where each cluster is partitioned into two or more clusters, and so on. This adds structure to flat clustering, giving an algorithm the ability to depict data similarity at different resolutions as well as an ancestral relationship between data points, as in the phylogenetic tree Kraskov et al. [2003].

On top of computational biology, hierarchical clustering has found various uses across computer imaging [Chen et al., 2021b, Selvan et al., 2005], computer security [Chen et al., 2020, 2021a], natural language processing [Ramanath et al., 2013], and much more. Moreover, it can be applied to any flat clustering problem where the number of desired clusters is not given. Specifically, a hierarchical clustering can be viewed as a structure of clusterings at different resolutions that all agree with each other (i.e., two points clustered together in a higher resolution clustering will also be together in a lower resolution clustering). Generally, hierarchical clustering techniques are quite impactful on modern technology, and it is important to guarantee they are both effective and unharmful.

Researchers have recognized the harmful biases unchecked machine learning programs pose. A few examples depicting racial discrimination include allocation of health care [Ledford, 2019], presentation of ads suggestive of arrest records [Sweeney, 2013], prediction of hiring success [Bogen and Rieke, 2018], and estimation of recidivism risk [Angwin et al., 2016]. A popular solution that has been extensively studied in the past decade is *fair machine learning*. Here, fairness concerns the mitigation of bias, particularly against protected classes. Most often, fairness is an additional constraint on the allowed solution space; we optimize for problems in light of this constraint. For instance, the notion of *individual fairness* introduced by the foundational work of Dwork et al. [2012] deems that an output must guarantee that any two individuals who are similar are classified similarly.

37th Conference on Neural Information Processing Systems (NeurIPS 2023).

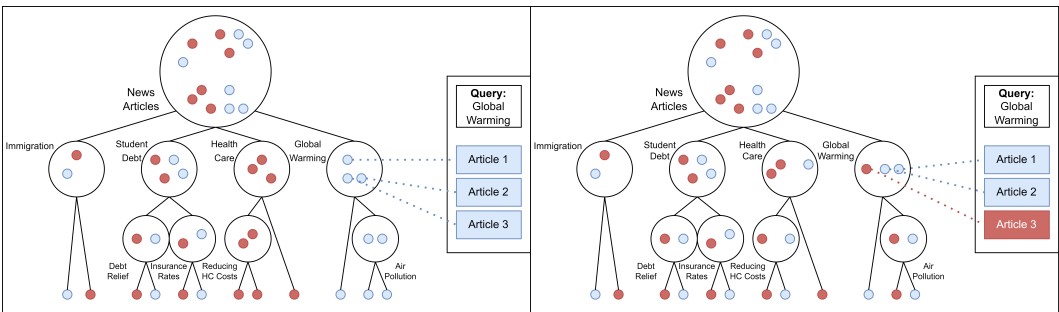

Figure 1: A hierarchical clustering of news articles. Red articles are conservative, blue are liberal. On the left is the optimal unfair hierarchy. We alter the hierarchy slightly on the right to achieve fairness. Now, the user's query for global warming will yield both liberal and conservative articles.

In line with previous work in clustering and hierarchical clustering, this paper utilizes the notion of *group fairness*, which enforces that different protected classes receive a proportionally similar distribution of classifications (in our case, cluster placement). Chierichetti et al. [2017] first introduced this as a constraint for the flat clustering problem, arguing that it mitigates a system's disparate impact, or non-proportional impact on different demographics. This notion of fair clustering has been similarly leveraged and extended by a vast range of works in both flat [Ahmadian et al., 2019, Bera et al., 2019, Bercea et al., 2019] and hierarchical [Ahmadian et al., 2020, Knittel et al., 2023] clustering.

To illustrate our fairness concept, consider the following application (Figure 1): a news database is structured as a hierarchical clustering of search terms, where a search term is associated with a cluster of news articles to output to the reader, and more specific search terms access finer-resolution clusters. When a user searches for a term, we simply identify the corresponding cluster and output the contained articles. However, as is, the system does not account for the political skew of articles. In Figure 1, we label conservative-leaning articles in red and liberal-leaning articles in blue. We can see that in this example, when the user searches for global warming articles, they will only see liberal articles. To resolve this, we add a group fairness constraint on our cluster: for example, require at least 1/3 of the articles in each cluster to be of each political skew. This guarantees (as depicted on the right) that the outputted articles will always be at least 1/3 liberal and 1/3 conservative, thus guaranteeing the user is exposed to a range of perspectives along this political axis. This notion of fairness, which we formally define in Definition 3, has been explored in the context of hierarchical clustering in both Ahmadian et al. [2020] and Knittel et al. [2023].

This paper is concerned with approximations for fair low-*cost* hierarchical clustering. Cost, first defined by Dasgupta [2016] and formally stated in Definition 2, is perhaps the most natural and well-motivated optimization metric for hierarchical clustering evaluation. Effectively, every pair of points contributes an additive cost of $similarity \times clust\_size$, where the latter is the size of the smallest cluster in the hierarchy containing both points. Unfortunately, it is quite difficult to optimize for (the best being $O(\sqrt{\log n})$-approximations [Charikar and Chatziafratis, 2017, Dasgupta, 2016]; it hypothesized to not be $O(1)$-approximable [Charikar and Chatziafratis, 2017]). This appears to be even more difficult in the hierarchical clustering literature. The first work to attempt this problem, Ahmadian et al. [2020], achieved a highly impractical $O(n^{5/6} \log^{3/2} n)$-approximation (not too far from the trivial $O(n)$ upper bound), posing fair low-cost hierarchical clustering as an interesting and inherently difficult problem. Knittel et al. [2023] greatly improved this to a near-polylog approximation factor of $O(n^\delta \text{polylog}(n))$, where $\delta$ can be arbitrarily close to 0, and parameterizes a trade-off between approximation factor and degree of fairness. Still, a true polylog approximation was left as an open problem, one which we solve in this paper.

## 1.1 Our Contributions

This work proposes the first polylogarithmic approximation for fair, low-cost hierarchical clustering. We leverage the work of Knittel et al. [2023] as a starting inspiration and create something much simpler, more direct, and better in both fairness and approximation. Like their algorithm, our algorithm starts with a low-cost unfair hierarchical clustering and then alters it with multiple well-defined and limited tree operators. This gives it a degree of explainability, in that the user can understand exactly the steps the algorithm took to achieve its result and why. In addition, our

algorithm achieves both relative cluster balance (i.e., clusters who are children of the same cluster have similar size) and fairness, along with a parameterizeable trade-off between the two.

On top of the benefits of Knittel et al. [2023]'s techniques, we propose a greatly simplified algorithm. They initially proposed an algorithm that required four tree operators, however, we only require two of the four, and we greatly simplify the more complicated operator. This makes the algorithm simpler to understand and more implementable. We show that even with this reduced functionality, we can cleverly achieve both a better approximation and degree of fairness:

**Theorem 1** (High-Level). *Given a $\gamma$-approximation to the cost objective for a hierarchical clustering on set of $n$ points, Algorithm 2 yields an $O(\log^2 n)$ approximation to cost which is relatively fair in time $O(n \log^2 n)$. Moreover, all clusters are $O(1/\log n)$ balanced, i.e., any cluster's children clusters are within $O(1/\log n)$ size different.*

To put this in perspective, previously, the best approximation for fair hierarchical clustering previously was $O(n^\delta \text{polylog}(n))$, whereas the best unfair approximation is $O(\sqrt{\log n})$. Our work greatly reduces this gap by providing a true $O(\text{polylog}(n))$ approximation, moreover with a low degree of 2. Note that this is not the most general and rigorous form of our result–for that, we defer to Theorem 1 as stated in the body of our paper. This simpler version is found by setting $k = O(1)$, $h = O(\log n)$, and $\epsilon = O(1/\log n)$ in the more general theorem (note we assume $\lambda = O(1)$).

## 2 Preliminaries

### 2.1 The Vanilla Problem

Fair clustering literature refers to the original problem variant, without fairness, as the "vanilla" problem. We define the vanilla problem of finding a low-cost hierarchical clustering here using our specific notation.

In this problem, we are given a complete graph $G = (V, E, w)$ with a weight function $w : E \rightarrow \mathbb{R}^+$ is a measure of the similarity between datapoints. Note the data is encoded as a complete tree because we require knowledge of all point-point relationships. We must construct a hierarchical clustering, represented by its dendrogram, $T$, with root denoted $\text{root}(T)$. $T$ is a tree with vertices corresponding to all clusters of the hierarchical clustering. Leaves of $T$, denoted $\text{leaves}(\text{root}(T))$ correspond to the points in the dataset (i.e., singleton clusters). An internal node $v$ corresponds to the cluster containing all leaf-data of the maximal subtree (i.e., contains all its descendants) rooted at $v$, $T[v]$. In addition, we let $u \wedge v$ denote the lowest common ancestor of $u$ and $v$ in $T$.

In order to define Dasgupta [2016]'s cost function, we use the same notational simplifications as Knittel et al. [2023]. For an edge $e = (x, y) \in E$, we say $n_T(e) = |\text{leaves}(T[x \wedge y])|$ is the size of the smallest cluster in the hierarchy containing $e$. Similarly, for a hierarchy node $v$, $n_T(v_i) = |\text{leaves}(T[v_i])|$ is the size of the corresponding cluster. This is sufficient to introduce the notion of *cost*.

**Definition 1** (Knittel et al. [2023]). *The **cost** of $e \in E$ in a graph $G = (V, E, w)$ in a hierarchy $T$ is* $\text{cost}_T(e) = w(e) \cdot n_T(e)$.

Dasgupta's cost function can then be written as a sum over the costs of all edges.

**Definition 2** (Dasgupta [2016]). *The **cost** of a hierarchy $T$ on graph $G = (V, E, w)$ is:*

$$\text{cost}(T) = \sum_{e \in E} \text{cost}_T(e)$$

Our algorithm begins by assuming we have some approximate vanilla hierarchy, $T$. That is, if $OPT$ is the optimal hierarchy tree, then $\text{cost}(T) \leq \alpha \cdot \text{cost}(OPT)$ for some approximation factor $\alpha$. According to Dasgupta [2016], we can transform this hierarchy to be binary without increasing the cost. Our paper simply assumes our input is binary. We then produce a modified hierarchy $T'$ which similar structure to $T$ that guarantees fairness, i.e., $\text{cost}(T') \leq \alpha' \cdot \text{cost}(OPT)$ for some approximation factor $\alpha' \geq \alpha$. Note this comparison is being made to the vanilla $OPT$, as we are unsure, at this time, how to classify the optimal fair hierarchy. Note that the binary assumption may not hold when we consider adding a fairness constraint.

## 2.2 Fairness and Balance Constraints

We consider the fairness constraints based off those introduced by Chierichetti et al. [2017] and extended by Bercea et al. [2019]. On a graph $G$ with colored vertices, let $\ell(C)$ count the number of $\ell$-colored points in cluster $C$.

**Definition 3** (Knittel et al. [2023]). *Consider a graph $G = (V, E, w)$ with vertices colored one of $\lambda$ colors, and two vectors of parameters $\alpha, \beta \in (0,1)^\lambda$ with $\alpha_\ell \leq \beta_\ell$ for all $\ell \in [\lambda]$. A hierarchy $T$ on $G$ is **fair** if for any non-singleton cluster $C$ in $T$ and for every $\ell \in [\lambda]$, $\alpha_\ell |C| \leq \ell(C) \leq \beta_\ell |C|$. Additionally, any cluster with a leaf child has only leaf children.*

Effectively, we are given bounds $\alpha_\ell$ and $\beta_\ell$ for each color $\ell$. Every non-singleton cluster must have at least an $\alpha_\ell$ fraction and at most a $\beta_\ell$ fraction of color $\ell$. This guarantees proportional representational fairness of each color in each cluster.

As an intermediate step in achieving fairness, we will create splits in our hierarchy that achieve relative balance in terms of subcluster size. Thus, the following definition will come in handy.

**Definition 4.** *In a hierarchy, a vertex $v$ (corresponding to cluster $C$) with $c_v$ children is $\epsilon$-**relatively balanced** if for every cluster $\{C_i\}_{i \in [c_v]}$ that corresponds to a child of $v$, $(\frac{1}{c_v} - \epsilon)|C| \leq |C_i| \leq (\frac{1}{c_v} + \epsilon)|C|$.*

While this definition is quite similar to that from Knittel et al. [2023], it deviates in two ways: 1) we only define it on a single split as opposed to the entire hierarchy and 2) we allow splits to be non-binary. If we apply it to the entire hierarchy and constrain it to be binary, it is equivalent to the former definition.

## 2.3 Tree Operators

Our work simplifies the work of Knittel et al. [2023]. In doing so, we follow the same framework, using tree operators to make well-defined and limited alterations to a given hierarchical clustering (Figure 2). In addition, our algorithm simplifies operator use in two ways: 1) we only utilize two of their four tree operators, and 2) we greatly simplified their most complicated operator and show that it can still be used to create a fair hierarchy.

Figure 2: Our operators: subtree deletion and insertion and shallow tree folding.

First off, we utilize the same subtree deletion and insertion operator. The main difference is how we use it, which will be discussed in Section 3. At a high level, this operator removes a subtree from one part of the hierarchy and reinserts it elsewhere, adding and removing parent vertices as necessary.

**Definition 5** (Knittel et al. [2023]). *Consider a binary tree $T$ with internal nodes $u$, some non-ancestor $v$, $u$'s sibling $s$, and $v$'s parent $g$. **Subtree deletion** at $u$ removes $T[u]$ from $T$ and contracts $s$ into its parent. **Subtree insertion** of $T[u]$ at $v$ inserts a new parent $p$ between $v$ and $g$ and adds $u$ as a second child of $p$. The operator $\mathrm{del\_ins}(u, v)$ deletes $u$ and inserts $T[u]$ at $v$.*

The other operator we leverage is their tree folding operator, however we greatly simplify it. In the previous work, tree folding took two or more isomorphic trees and mapped the internal nodes to each other. Instead, we simply take two or more subtrees and merge their roots. The new root then directly splits into all children of the roots of all folded trees. In a way, this is an implementation of their folding operator but only at a single vertex in the tree topology. This is why we call it a shallow tree fold.

**Definition 6.** *Consider a set of subtrees $T_1, \ldots, T_k$ of $T$ such that all $\mathrm{root}(T_i)$ have the same parent $p$ in $T$. A **shallow tree folding** of trees $T_1, \ldots, T_k$ ($\mathrm{shallow\_fold}(T_1, \ldots, T_k)$) modifies $T$ such that all $T_1, \ldots, T_k$ are replaced by a single tree $T_f$ whose root $\mathrm{root}(T_f)$ is made a child of $p$, and whose root-children are $\cup_{i \in [k]} \mathrm{children}(\mathrm{root}(T_i))$.*

In addition, we assume the subtree $T_f$ is then arbitrarily binarized [Dasgupta, 2016] after folding. Since our algorithm works top-bottom, creating balanced vertices as it goes, we don't yet care about the fairness of the descendants of $T_f$. Moreover, we will then recursively call our algorithm on $T_f$ to do precisely this.

# 3 Main Algorithm

In this section, we present our fair, low-cost, hierarchical clustering algorithm along with its analysis. Ultimately, we achieve the following (for a more intuitive explanation, see Section 1):

**Theorem 1.** *When $T$ is a $\gamma$-approximate low-cost vanilla hierarchical clustering over $\ell(V) = c_\ell n = O(n)$ vertices of each color $\ell \in [\lambda]$, MakeFair (Algorithm 2), for any constants $\epsilon, h, k$ with $h \gg k^\lambda$ and $n \gg h$, runs in $O(n \log n(h + \lambda \log n))$ time and yields a hierarchy $T'$ satisfying:*

1. *$T'$ is an $O\left(\frac{(h-1)}{\epsilon} + \frac{1+\epsilon}{1-\epsilon}k^\lambda\right) \gamma$-approximation for cost.*

2. *$T'$ is fair for any parameters for all $i \in [\lambda]$: $\alpha_i \leq \frac{\lambda_i}{n}\left(\frac{1-\epsilon}{(1+\epsilon)^2}\left(1 - \frac{k(1+\epsilon)}{c_i h}\right)\right)^{O(\log(n))}$ and*

   *$\beta_i \geq \frac{\lambda_i}{n}\left(\frac{1+\epsilon}{(1-\epsilon)^2}\left(1 + \frac{1-\epsilon}{c_i k}\right)\right)^{O(\log(n))}$, where $\lambda_i = c_i n$.*

3. *All internal nodes in $T'$ are $\epsilon$-relatively balanced.*

The main idea of our algorithm is to leverage similar tree operators to that of Knittel et al. [2023], but greatly simplify their usage and apply them in a more direct, careful manner. Specifically, the previous work processes the tree four times: once to achieve $1/6$-relative balance everywhere, next to achieve $\epsilon$-relative balance, next to remove the bottom of the hierarchy, and finally to achieve fairness. The problem is that this causes proportional cost increases to grow in an exponential manner, particularly because the relative balance significantly degrades as you descend the hierarchy. Our solution is to instead do a single top to bottom pass of the tree, rebalancing and folding to achieve fairness as we go. We describe this in detail now.

First, we assume our input is some given hierarchical clustering tree. Ideally, this will be a good approximation for the vanilla problem, but our results do work as a black box on top of any hierarchical clustering algorithm. Second, we apply SplitRoot in order to balance the root (Section 3.1). And finally, we apply shallow tree folding on the children of the root to achieve fairness (Section 3.2). This gives us the first layer of our output, and then we recurse.

## 3.1 Root Splitting and Balancing

SplitRoot is depicted in Algorithm 1. This fills the role of Knittel et al. [2023]'s Refine Rebalance Tree algorithm (and skips their Rebalance Tree algorithm), but it functions differently in that it only rebalances the root and it immediately splits the root into $h$ children, according to our input parameter $h$.

We start SplitRoot by adding dummy children to $v$ until it has $h$ children (recall we can assume the input is binary). A dummy or null child is just a placeholder for a child to be constructed, or alternatively simply a zero-sized tree (note: this does not add any leaves to the tree). None of these children will be left empty in the end. Next, we define $v_{max}$ and $v_{min}$, the maximal subtrees rooted at $\text{children}(\text{root}(T'))$ which have the most and fewest leaves, respectively.

As long as the root is not $\epsilon$-relatively balanced (which is equivalent to $n_{T'}(v_{max})$ or $n_{T'}(v_{min})$ deviating from the target $n/h$ by over $n\epsilon$, as they are extreme points), we will attempt to rebalance. We define $\delta_1$ and $\delta_2$ to be the proportional deviation of $n_{T'}(v_{min})$ and $n_{T'}(v_{max})$ from the target size $n/h$ respectively, and $\delta$ to be the minimum of the two. In effect, $\delta$ measures the maximum number of leaves we can move from the large subtree to the small subtree without causing $n_{T'}(v_{max})$ to dip below $n/h$ or $n_{T'}(v_{min})$ to peak above $n/h$. This is important to guarantee our runtime: as an accounting scheme, we show that clusters monotonically approach size $n/h$, and thus we can quantify how fast our algorithm completes. We fully analyze this later, in Lemma 2.

Now we must attempt exactly this procedure: move a large subtree from $v_{max}$ to $v_{min}$, though this subtree can have no more than $\delta n$ leaves. To do this, we simply start at $v_{max}$ and traverse down its right children (recall below $v_{max}$, the tree is still binary). We halt on the first child that is of size $\delta n$ or smaller. We then remove it and find a place to reinsert it under $v_{min}$.

The insertion spot is found similarly by descending down $v_{min}$'s left children until the right child of the current vertex has fewer leaves in its subtree than the tree we are inserting. Thus, we have

completed our insertion and deletion operations. We repeat until the tree is relatively balanced as desired.

---

**Algorithm 1** SplitRoot

---

**Input:** A binary hierarchy tree $T$ of size $n \geq 1/2\epsilon$ over a graph $G = (V, E, w)$, with smaller cluster always on the left, and parameters $h \in [n]$ and $\epsilon \in (0, \min(1/6, 1/h))$.
**Output:** A hierarchical clustering $T'$ with an $\epsilon$-relatively balanced root that has $k$ children.
1: Initialize $T' = T$
2: $v = \text{root}(T')$
3: Add null children to $v$ until it has $h$ children
4: Let $v_{min} = \text{argmin}_{v' \in \text{children}(v)} n_{T'}(v')$
5: Let $v_{max} = \text{argmax}_{v' \in \text{children}(v)} n_{T'}(v')$
6: **while** $n_{T'}(v_{max}) > n(1/h + \epsilon)$ or $n_{T'}(v_{min}) < n(1/h - \epsilon)$ **do**
7: $\quad \delta_1 = 1/h - n_{T'}(v_{min})/n$
8: $\quad \delta_2 = n_{T'}(v_{max})/n - 1/h$
9: $\quad \delta = \min(\delta_1, \delta_2)$
10: $\quad$ Let $v = v_{max}$
11:
12: $\quad$ **while** $n_{T'}(v) > \delta n$ **do**
13: $\quad\quad v \leftarrow \text{right}_{T'}(v)$
14: $\quad$ **end while**
15:
16: $\quad u \leftarrow v_{min}$
17: $\quad$ **while** $n_{T'}(\text{right}_{T'}(u)) \geq n_{T'}(v)$ **do**
18: $\quad\quad u \leftarrow \text{left}_{T'}(u)$
19: $\quad$ **end while**
20: $\quad T' \leftarrow T'.\text{del\_ins}(u, v)$
21: $\quad$ Reset $v_{min}$ and $v_{max}$
22: **end while**

---

We now analyze this part of the algorithm. The full proofs can be found in Appendix B, and we proceed to give the intuition here. To start, consider the tree we are deleting and reinserting, $T'[v]$. Ideally, we want this to have many leaves, but no more than $\delta n$. We demonstrate that:

**Lemma 1.** *For a subtree $T'[v]$ that is deleted and reinserted in* SplitRoot *(Algorithm 1), we must have that $\epsilon n/(2(h-1)) < n_T(v) \leq \delta n$.*

The upper bound simply comes from our stopping condition in the first nested while loop: we ensure $n_{T'}(v) \leq \delta n$ before selecting it. The lower bound is slightly more complicated. Effectively, we start by noting that $\max(\delta_1, \delta_2) > \epsilon$, because otherwise the stopping condition for the outer loop would be met. Then, consider the total amount of "excess of large clusters", or more precisely, the sum over all deviations from $n/h$ of clusters larger than $n/h$ (note if all clusters were $n/h$, it would be perfectly balanced). This total excess must be matched in the "deficiency of small clusters", which is the sum of deviations of clusters smaller than $n/h$. Therefore, since there are at least $h$ small or $h$ large clusters, the largest deviation must be at most $h$ times the smallest deviation, according to our accounting scheme. This allows us to bound $\delta \geq \epsilon/(h-1)$. The tree that is inserted and deleted must have at least half this many leaves, since it is the larger child of a node with over $\delta n$ leaves in its subtree. This gives our lower bound, showing we move at least a significant number of vertices each step.

Next, we aim to illustrate the relative balance. In addition to our analysis, we also obtain the runtime, which exhibits near-linearity when the condition $h \ll n$ holds.

**Lemma 2.** SplitRoot *(Algorithm 1) yields a hierarchy whose root is $\epsilon$-relatively balanced with $h$ children. In addition, it requires $O(nh)$ time to terminate.*

The root has $h$ children by definition at the algorithm start and this remains invariant. The runtime comes from our aforementioned accounting scheme: the total excess and deficiency is reduced by the number of leaves in the subtree we move at each step, which we showed in Lemma 1 is $n\epsilon/(2(h-1))$ at least. This gives us a convergence time of $O(h)$, and each step can be bounded by $O(n)$ time as we search for our insertion and deletion spots. Finally, the balance comes from the fact that our stopping condition is equivalent to the root being relatively balanced.

All that remains is to show the negative impact on the cost of edges that are separated by the algorithm. We bound this via the following lemma.

**Lemma 3.** *In* SplitRoot *(Algorithm 1), for all $e \in E$ that are separated:*

$$\text{cost}_{T'}(e) \leq n \cdot w(e) \leq 2(h-1) \cdot \text{cost}_T(e)/\epsilon$$

Lemma 1 gives us that moved subtrees are at least of size $\epsilon n/(2(h-1))$, which is a lower bound on the size of the smallest cluster containing any edge separated by the algorithm. This is due to the fact that separated edges must have one endpoint in the deleted subtree and one outside, so their least common ancestor is an ancestor of the subtree. At worst, the final size of the smallest cluster containing such an edge is $n$, so the proportional increase is $2(h-1)/\epsilon$ at worst.

## 3.2 Fair Tree Folding

Next, we discuss how to achieve fairness by using MakeFair, as seen in Algorithm 2. This is our final recursive algorithm which utilizes SplitRoot. Assume we are given some hierarchical clustering. We start by running SplitRoot, to balance the split at the root and give it $h$ children. Next we use a folding process similar to that of Knittel et al. [2023], but we use our shallow tree fold operator.

More specifically, we first sort the children of the root by the proportional representation of the first color (say, red). Then, we do a shallow fold across various $k$-sized sets, defined as follows: according to our ordering over the children, partition the vertices into $k$ contiguous chunks starting from the first vertex. For each $i \in [h/k]$, we find the $i$-th vertex in each chunk and fold them together. Notice that this is a $k$-wise fold since there are $k$ chunks, and we end up with $h/k$ vertices. This is repeated on each color. After this, we simply recurse on the children. If a child is too small to be balanced by SplitRoot, then we stop and give it a trivial topology (a root with many leaf-children).

This completes our algorithm description. We now evaluate its runtime, degree of fairness, and approximation factor. To start, we show the degree of fairness achieved at the top level of the hierarchy.

**Lemma 4.** MakeFair *(Algorithm 2) yields a hierarchy such that all depth 1 vertices satisfy fairness under $\alpha_i \leq \frac{\lambda_i}{n} \cdot \frac{1-\epsilon}{(1+\epsilon)^2} \left(1 - \frac{k(1+\epsilon)}{c_i h}\right)$ and $\beta_i \geq \frac{\lambda_i}{n} \cdot \frac{1+\epsilon}{(1-\epsilon)^2} \left(1 + \frac{1-\epsilon}{c_i k}\right)$, where $\lambda_i = c_i n$.*

This proof is quite in depth, and most details are deferred to Appendix B. At a high level, we are showing that the folding process guarantees a level of fairness. The parts in our partition are ordered by the density of the color (say, red). Since each final vertex is made by folding across one vertex in each part, meaning that the vertices have a relatively wide spread in terms of their density of red points. This means that red vertices are distributed relatively well across our final subtrees. This guarantees a degree of balance.

The problem is that the degree of fairness still exhibits a compounding affect as we recurse. More specifically, since the first children are not perfectly balanced, then in the next recursive step, the total data subset we are working on may now deviate from the true color proportions. This deviation is bounded by our result in Lemma 4, but it will increase proportionally at each step.

**Lemma 5.** *In* MakeFair *(Algorithm 2), let $\{\lambda_i\}_{i \in [\lambda]}$ be the proportion of each color and assume $k^\lambda \ll h$. At any recursive call, the proportion of any color is (where $\lambda_i = 1/c_i$ for constant $c_i$):*

$$\lambda_i \left(\frac{1-\epsilon}{(1+\epsilon)^2}\left(1 - \frac{k(1+\epsilon)}{c_i h}\right)\right)^{O(\log(n/h))} \leq \lambda_i^j \leq \lambda_i \left(\frac{1+\epsilon}{(1-\epsilon)^2}\left(1 + \frac{1-\epsilon}{c_i k}\right)\right)^{O(\log(n/h))}$$

*Moreover, the recursive depth is bounded above by $O(\log(n/h))$.*

This result derives directly from Lemma 4. Effectively, we increase the proportion of each color by the same factor each recursive step. All that is left to do is bound the recursive depth. Notice we start with $n$ vertices. After splitting, our subtrees have size at most $(1+\epsilon)n/h$. After one fold, this is increased by a factor of $k$, and thus $k^\lambda$ after all folds. Interestingly, this doesn't impact the final result significantly; it's fairly similar to turning an $n$-sized tree into an $n/h$-sized tree, giving an $O(\log(n/h))$ recursive depth. This will be sufficient to show our fairness.

Next, we evaluate the cost incurred at each stage in the hierarchy.

**Lemma 6.** *In* MakeFair *(Algorithm 2), for all $e \in E$ that is separated before the recursive call:*

$$\text{cost}_{T'}(e) \leq O\left(\frac{2(h-1)}{\epsilon} + \frac{1+\epsilon}{1-\epsilon}k^\lambda\right)\text{cost}_T(e)$$

---

**Algorithm 2** MakeFair

---

**Input:** A hierarchy tree $T$ of size $n \geq 1/2\epsilon$ over a graph $G = (V, E, w)$ with vertices given one of $\lambda$ colors, and parameters $h \in [n]$, $k \in [h/(\lambda - 1)]$, and $\epsilon \in (0, \min(1/6, 1/h))$.
**Output:** A fair hierarchical clustering $T'$.
 1: $T' = \text{SplitRoot}(T, h, \epsilon)$
 2: $h' \leftarrow h$
 3: **for** each color $\ell \in [\lambda]$ **do**
 4:     Order $\{v_i\}_{i \in [h']} = \text{children}(\text{root}(T'))$ decreasing by $\frac{\ell(\text{leaves}(v_i))}{n_{T'}(v_i)}$
 5:     For all $i \in [k]$, $T' \leftarrow T'.\text{shallow\_fold}(\{T'[v_{i+(j-1)k}] : j \in [h'/k]\})$
 6:     $h' \leftarrow h'/k$
 7: **end for**
 8: **for** each child $v_i$ of $\text{root}(T')$ **do**
 9:     **if** $n \geq \max(1/2\epsilon, h)$ **then**
10:         Replace $T'[v_i] \leftarrow \text{MakeFair}(T'[v_i], h, k, \epsilon)$
11:     **else**
12:         Replace $T'[v_i]$ with a tree of root $v_i$, leaves $\text{leaves}(T'[v_i])$, and depth 1.
13:     **end if**
14: **end for**

---

As discussed before, the final cluster size should be $(1 + \epsilon)nk^\lambda/h$. Any separated edge must have a starting cluster size of at least $(1 - \epsilon)n/h$, as this is the size of the smallest cluster involved in tree folding. From this, it is straightforward to compute the proportional cost increase of a single recursive level, as well as the cost increase from the initial splitting in Lemma 3.

We additionally demonstrate that, whenever an edge is separated, its endpoints' least common ancestor will no longer be involved in any further recursive step. More formally:

**Lemma 7.** *In* MakeFair *(Algorithm 2), any edge $e \in E$ is separated at only one level of recursion.*

Putting these two together pretty directly gives us our cost approximation.

**Lemma 8.** *In* MakeFair *(Algorithm 2),* $\text{cost}(T') \leq O\left(\frac{2(h-1)}{\epsilon} + \frac{1+\epsilon}{1-\epsilon}k^\lambda\right)\text{cost}(T)$.

Finally, Theorem 1 comes directly from Lemmas 6 and 8.

## 4 Simulations

This section validates the theoretical guarantees of Algorithm 2. Specifically, we demonstrate that modifying an unfair hierarchical clustering using the presented procedure yields a fair hierarchy that incurs only a modest increase in cost.

**Datasets.** We use two data sets, *Census* and *Bank*, from the UCI data repository Dua and Graff [2017]. Within each, we subsample only the features with numerical values. To compute the *cost* of a hierarchical clustering we set the similarity to be $w(i, j) = \frac{1}{1+d(i,j)}$ where $d(i, j)$ is the Euclidean distance between points $i$ and $j$. We color data based on binary (represented as blue and red) protected features: *race* for *Census* and *marital status* for *Bank* (both in line with the prior work of Ahmadian et al. [2020]). As a result, *Census* has a blue to red ratio of 1:7 while *Bank* has 1:3. We then subsample each color in each data set such that we retain (approximately) the data's original balance. We use samples of size 512 for the balance experiments, and vary the sample sizes when assessing cost. For each experiment we conduct 10 independent replications (with different random seeds for the subsampling), and report the average results. We vary the parameters $(c, h, k, \varepsilon)$[1] to experimentally assess their theoretical impact on the approximate guarantees of Section 3. Due to space constrains, we here present only the results for the *Census* dataset and defer the complimentary results on *Bank* to the appendix.

---

[1]Note that the $c$ parameter is used to scale $\varepsilon$.

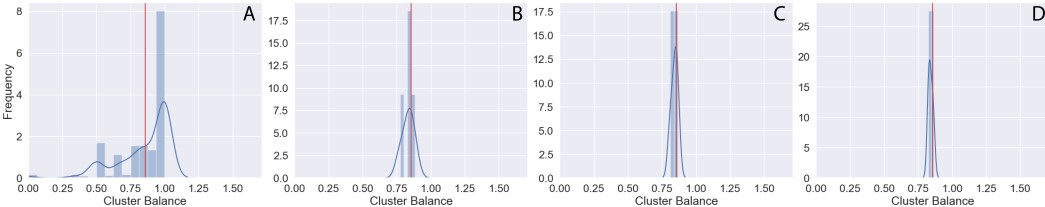

Figure 3: Histogram of cluster balances after tree manipulation by Algorithm 2 on a subsample from the *Census* dataset of size $n = 512$. The four panels depict: **(A)** cluster balances after applying the (unfair) average-linkage algorithm, **(B)** the resultant cluster balances after running Algorithm 2 with parameters $(c, h, k, \varepsilon) = (8, 4, 2, 1/c \cdot \log_2 n)$, **(C)** cluster balances after tuning $c = 4$, **(D)** cluster balances after further tuning $c = 2$. The vertical red line on each plot indicates the balance of the dataset itself.

**Implementation.** The Python code for the following experiments are available in the Supplementary Material. We start by running average-linkage, a popular hierarchical clustering algorithm. We then apply Algorithm 2 to modify this structure and induce a *fair* hierarchical clustering that exhibits a mild increase in the cost objective.

**Metrics.** In our results we track the approximate cost objective increase as follows: Let $G$ be our given graph, $T$ be average-linkage's output, and $T'$ be Algorithm 2's output. We then measure the ratio $\text{RATIO}_{cost} = cost_G(T')/cost_G(T)$. We additionally quantify the fairness that results from application of our algorithm by reporting the balances of each cluster in the final hierarchical clustering, where true fairness would match the color proportions of the underlying dataset.

**Results.** We first demonstrate how our algorithm adapts an unfair hierarchy into one that achieves fair representation of the protected attributes as desired in the original problem formulation.

In Figure 3, we depict the cluster balances of an *unfair* hierarchical clustering algorithm, namely "average-linkage", and subsequently demonstrate that our algorithm effectively concentrates all clusters around the underlying data balance. In particular, we first apply the algorithm and then show how we the balance is further refined by tuning the parameters. The application of Algorithm 2 dramatically improves the representation of the protected attributes in the final clustering and, as such, firmly resolves the problem of achieving fairness.

While reaching this fair partitioning of the data is the overall goal, we further demonstrate that, in modifying the unfair clustering, we only increase the cost approximation by a modest amount. Figure 4 illustrates the change in relative cost as we increase the sample size $n$, the primary influence on our theoretical cost guarantees of Section 3. Specifically, we vary $n$ in $\{128, 256, 512, 1024, 2048\}$ and compute 10 replications (on different random seeds) of the fair hierarchical clustering procedure. Figure 4 depicts the mean relative cost of these replications with standard error bars. Notably, we see that the cost does increase with $n$ as expected, but the increase relative to the unfair cost obtain by average linkage is only by a small multiplicative factor.

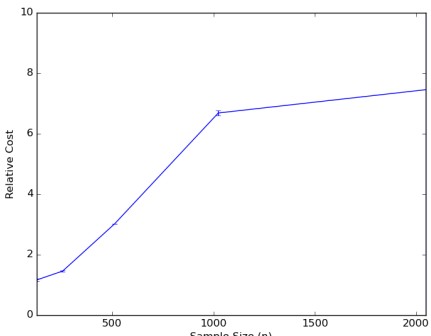

Figure 4: Relative cost of the fair hierarchical clustering resulting from Algorithm 2 compared to the unfair clustering as a function of the sample size $n$.

As demonstrated through this experimentation, the simplistic procedure of Algorithm 2 not only ensures the desired fairness properties absent in conventional (unfair) clustering algorithms but accomplishes this feat with a negligible rise in the overall cost. These results further highlight the immense value of our work.

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
