# OpenReview forum: "Fair, Polylog-Approximate Low-Cost Hierarchical Clustering"
_NeurIPS.cc/2023/Conference — NeurIPS 2023 poster_

### Official Review · Reviewer_BAkf · 2023-07-06

**Soundness:** 3 good
**Presentation:** 2 fair
**Contribution:** 3 good
**Rating:** 6
**Confidence:** 4

**Summary:**

This paper studies the clustering problem in a fair setting. It proposes an approximation algorithm that achieves polylogarithmic factors for fair and cost while also keep relative balance. This work has greatly improved the result of Knittel et al. [2023] that has an approximation factor of $O(n^\delta\text{polylog}(n))$. The simulation results on two datasets verifies the effectiveness of their algorithm (replacing the binary clustering algorithm that theoretically achieves a factor of $O(\sqrt{\log n})$ with average-linkage).

**Strengths:**

(1) A theoretical result of an approximation algorithm for the fair clustering problem, with great improvement in approximation factor.

(2) Simple and intuitive tree operations in clustering adjustment that can be theoretically analysed.

**Weaknesses:**

(1) This work seems to heavily depend on Knittel et al. [2023], although the authors are probably overlapping heavily.

(2) Scalability of $h$, $k$ and $\lambda$ is doubtful. According to Theorem 1, $h$ should be much larger than $k^\lambda$, but in the experiments, $h=4$, $k=\lambda=2$ such that $h=k^\lambda$, which seems to be a violation of the settings.

(3) There are some unclear points (refer to the questions).

**Questions:**

(1) What is the role of $\gamma$ in Theorem 1? Do you mean an approximation algorithm for the cost of the input binary clustering tree (as described in Lemma 8) rather than the optimal cost of the original problem? So, for the latter, $\gamma$ needs to be multiplied outside the approximation factor, right?

(2) How do you define the cluster balance? What does $c$ mean in the simulation settings?

(3) Line 171, Def. 6, do you mean root$(T_f)$, rather than root$(T)$?

(4) Line 236, do you mean "at most $h$" rather than "at least $h$"?

(5) A typo: Line 281, "poitns" should be "points".

(6) A ref error: Line 581.

(7) Please delete Line 532 that has revealed author information!

**Limitations:**

Yes, the authors have addressed the limitations in a specific section (Appendix A).

---

> ### Author Rebuttal · Authors · 2023-08-09
>
> *“This work seems to depend heavily on Knittel 2023…”*
>
> This work does build upon the algorithm of Knittel 2023, but greatly simplifies the procedure to be more easily implemented. Moreover, our analysis removes the extra $n^\delta$ approximation factor which allows for the first polylogarithmic approximation to cost with fairness. Since the best known unfair cost approximation is also polylogarithmic, this result is substantial in showing that a clustering can be made fair with only a modest increase in the approximation factor. In a sense the “price of fairness” is demonstrated to be low.
>
> *“Scalability of $h, k$, and $\lambda$ is doubtful…”*
>
> While the runtime may not be technically scalable in terms of these parameters, these parameters are much less than $n$. In consideration of this, the runtime is certainly scalable in $n$. Since $n \gg h \gg \lambda, k,$ this will certainly be $O(n^2 \log(n))$. Notice that for our Corollary, we actually assume the largest parameter is $h = O(\log n)$. Therefore, the runtime should be something like $O(n \log^2(n))$.
>
> *“What is the role of $\gamma$?”*
>
> As correctly noted, $\gamma$ is the approximation factor to cost incurred by the base unfair hierarchical clustering algorithm. And yes, gamma should be included in the approximation factor, we will correct this. To date, the best known such algorithm admits an $O(\sqrt{\log n})$ approximation factor.
>
> *“How do you define the cluster balance? What is c in the experiments?”*
>
> We apologize for the confusion in the presentation. The parameter noted as $c$ in the experiments is used to define the $\epsilon$ value and is not related to the notation for cluster balance in theoretical results. We will rectify this in the camera-ready version of our paper.
>
> We lastly thank the reviewer for identifying typos and other mistakes, which have now been corrected.

---

> > ### Comment · Reviewer_BAkf · 2023-08-18
> >
> > Thank the authors' response. The technical difference from Knittel 2023 is still unclear for me, even though after reading the rebuttal to Reviewer gEkD. Sorry for not having enough time to check all the details. More intuitive comparisons to Knittel 2023 are required, especially in the case that many operations are the same. I stick to my score.

---

### Official Review · Reviewer_XyRG · 2023-07-08

**Soundness:** 4 excellent
**Presentation:** 4 excellent
**Contribution:** 4 excellent
**Rating:** 8
**Confidence:** 2

**Summary:**

  The paper addresses the problem of fair hierarchical clustering.
Given a hierarchical clustering with a certain cost (determined by
Dasgupta's cost function), the paper shows how the hierarchy (tree) can be
efficiently modified so that fairness constraints are satisfied
and the cost of the tree increases (provably) by only by a
small factor. The theoretical results are validated through experiments
on two data sets.

**Strengths:**

(a) The paper provides the first (provable) polylog approximation
  for the cost of hierarchical clustering while satisfying fairness
  requirements. This is an important advance in fair hierarchical clustering.

  (b) The algorithm simplifies a previously known algorithm while
  achieving the approximation bound.

  (c) The paper is written very well.

**Weaknesses:**

  A (very) minor weakness is that the statements of theorems require
a good amount of effort to understand since they involve many different
parameters.

**Questions:**

  (1) On lines 124--126 (page 4), it is mentioned that the approximation
  factor is with respect to an optimal vanilla hierarchy (which doesn't
  consider fairness).  Did the previous work also use this assumption?

  (2) Are any hardness of approximation results known for the problem? (It will
  be interesting to see how close the polylog approximation is compared to
  what can't be achieved in polynomial time under a standard assumption
  such as P != NP.)

  (3) The statements of theorems/lemmas etc. involve many parameters. This
  reviewer agrees that this is unavoidable for making the theorems precise.
  Would it be possible to include informal statements of results that are easier
  to understand (even if they are not mathematically accurate)?

  Some minor typos:

  (a) Line 62 (page 2): "as a in" ---> "as an"
  (a) Line 121 (page 4): "increasing cost" ---> "increasing the cost"

---

> ### Author Rebuttal · Authors · 2023-08-09
>
> *“On lines 124--126 (page 4), it is mentioned that the approximation factor is with respect to an optimal vanilla hierarchy (which doesn't consider fairness). Did the previous work also use this assumption?”*
>
> Yes, the previous work also considers an approximation to an optimal, unfair, hierarchy, since it is currently not known how much worse an optimal fair hierarchy may be. One can simply view it as a lower bound on the optimal fair cost, however we note it in the results since it is a strictly stronger statement. It is interesting to note that the best known (unfair) cost approximation is also a power of log(n), specifically ½, so our result does not deviate too considerably from this. Though it would be nice to shoot for logarithmic or possibly sublogarithmic to narrow this gap. That's a good future direction.
>
> *“Are any hardness of approximation results known for the problem? (It will be interesting to see how close the polylog approximation is compared to what can't be achieved in polynomial time under a standard assumption such as P != NP.)”*
>
> The only known hardness results are for the cost optimization in the unfair setting (which is known to be APX hard). We acknowledge the interest in determining a lower bound on the approximation factor for fair clustering hardness. This, however, would require distinguishing the fair optimum to the unfair optimum. No previous work has managed to do that, in fact, all approximations for this problem (ours and previous) compared the output cost to the unfair optimum cost, which is a lower bound on the fair optimum cost. Differentiating between the two in any way would be a very useful result!
>
> The statements of theorems/lemmas etc. involve many parameters. This reviewer agrees that this is unavoidable for making the theorems precise. Would it be possible to include informal statements of results that are easier to understand (even if they are not mathematically accurate)?”
>
> We intend to incorporate the following theorem (sketches) in the main text to address these concerns and improve the overall readability of our results:
>
> *Given a $\gamma$-approximation to the cost objective for a hierarchical clustering on set of $n$ points, Algorithm 2 yields an $O(\log^2 n)$ approximation to cost which is fair in time $O(n \log^2 n)$. Moreover, all clusters are $O(1/\log n)$ balanced, ie. any cluster’s children clusters are within $O(1 / \log n)$ size different.*
>
> This result is highlighted in Corollary 1, but will be expanded upon with more intuition for the definitions before their formal definition in Section 2 to improve readability.

---

> > ### Comment · Reviewer_XyRG · 2023-08-10
> >
> > I have checked the rebuttal. My questions have been answered in a satisfactory manner.

---

### Official Review · Reviewer_qPF4 · 2023-07-12

**Soundness:** 3 good
**Presentation:** 2 fair
**Contribution:** 2 fair
**Rating:** 6
**Confidence:** 3

**Summary:**

The paper considers finding a hierarchical clustering of (approximately) minimum Dasgupta cost under fairness and balance constraints. It appears that for any constant fairness range (i.e. the quantities a_i and b_i) they have a quasi-linear time algorithm which has a polylogarithmic approximation ratio. In fact - their algorithm takes any (unfair) hierarchical clustering and converts it into a fair one.

**Strengths:**

The best previous polynomial-time algorithm for approximately minimising the Dasgupta cost under fairness constriants had a polynomial approximation ratio (with exponent dependent on the fairness range). I think that going from a polynomial to polylogarithmic approximation ratio is a big step.
The authors have conducted experiments on their algorithm - showing empirically that the cost of the fair clustering produced is not much greater than the cost of the original (unfair) clustering. However, there are no experiments comparing to other algorithms.

**Weaknesses:**

The algorithm is tailored to the Dasgupta cost which is only one measure of goodness of a hierarchical clustering and in itself can be seen as a heuristic. However - I believe this cost to be very famous so the result is significant.
The way the bound is written at the moment is completely impenetrable - please see the “Questions” section for advice on this.
There are issues with the writing of the paper which I shall now list (I include here any typos etc.):
- The Introduction section uses a lot of notation that is not defined until later in the paper. This should definitely be fixed.
- The authors state that the cost is hypothesised not to be O(1) approximatable. What they mean is not O(1) approximatable by a polynomial-time algorithm.
- Definition 6 is ambiguous and hence is not a proper definition.
- In line 60 I believe the authors mean “fairness literature”
- In line 84 \gamma must surely appear in the approximation ratio
- Line 103 should start with “which is” instead of “is”
- In line 103 “tree” should be replaced by “graph”
- The use of “i.e.” in line 123 is wrong.
- Line 161 should say “inserts a new vertex p…”
- In line 133 \ell(C) (which I assume is the number of elements of C of colour \ell) is not defined. This notation also overloads \ell - it is both a colour and a function (corresponding to the colour). I recommend something like N_{\ell}(C).

**Questions:**

Suppose I give you a<1 and b>1 and I enforce you to create a fair hierarchical clustering with a_i=a and b_i=b for all colours i (where a_i and b_i are as in line 95). Can you write the parameters, the time complexity and the approximation ratio in terms of a and b? This is not just a question - doing this will vastly improve the presentation of the result since this is the logical way round (I give you the fairness constraints and you give me the (approximately) best hierarchical clustering satisfying those constraints).

**Limitations:**

Authors have addressed limitations

---

> ### Author Rebuttal · Authors · 2023-08-09
>
> *“The way the bound is written is completely impenetrable…”*
>
> We emphasize, as pointed out by reviewer XyRG, that for the given problem, our results require some in depth math to be fully accurate. To alleviate these issues, we will provide informal definitions and theorem statements in the response to reviewer XyRG.
>
> *“Suppose I give you $a<1$ and $b>1$ and I enforce you to create a fair hierarchical clustering with $a_i=a$ and $b_i=b$ for all colours $i$ (where $a_i$ and $b_i$ are as in line 95). Can you write the parameters, the time complexity and the approximation ratio in terms of $a$ and $b$?”*
>
> We note that $b_i$ should always be $\le 1$ since it’s the upper bound on the fractional representation of a color in a cluster (so if $b_i > 1$, it’s equivalent to $b_i = 1$, since that is the “no upper bound” case). We will answer the question assuming $0 < a < b < 1$, and all $a_i = a$ and $b_i = b$. In that case, yes you could derive a set of parameters needed to achieve this degree of fairness in terms of $a$ and $b$, and then plug them into the time complexity and approximation ratios. However, since there is a tradeoff between our parameters $\epsilon, h$, and $k$, and how they affect the degree of fairness, there will actually be a large range of parameterizations one can work with, so we cannot pose a straightforward answer. Realistically, this might be best done programmatically with parameter tuning.
>
> We thank the reviewer for the suggested revisions to improve readability of our main results. To more intuitively present the results, we intend to include the following informal theorem statement in the introduction (and defer the full parametrization to the analysis):
>
> *Given a $\gamma$-approximation to the cost objective for a hierarchical clustering on set of $n$ points, Algorithm 2 yields an $O(\log^2 n)$ approximation to cost which is relatively fair in time $O(n \log^2 n)$. Moreover, all clusters are $O(1/\log n)$ balanced, ie. any cluster’s children clusters are within $O(1 / \log n)$ size different.*
>
> Where we state that “relative fairness” simply implies looser constraints on the proportional representation of each color in each cluster. Lastly, we thank the reviewer for noting typos and grammatical errors, all of which have  now been corrected.

---

> > ### Comment · Reviewer_qPF4 · 2023-08-16
> >
> > I have looked at the other reviews and the rebuttal and will not be changing my score. I very much like the result and I'm happy for this paper to be accepted - but if it is then it should be cleaned up (by incorporating the changes suggested by myself and the other reviewers) before publication. I would like to note that reading this paper inspired me to get into the field of fairness myself - thanks for that :-)

---

> > > ### Comment · Reviewer_qPF4 · 2023-08-16
> > >
> > > On further thought I've increased my score to 6

---

### Official Review · Reviewer_gEkD · 2023-07-22

**Soundness:** 2 fair
**Presentation:** 1 poor
**Contribution:** 2 fair
**Rating:** 3
**Confidence:** 3

**Summary:**

This paper considers fairness in hierarchical clustering and proposes an approximation algorithm that modifies a given unfair approximate vanilla hierarchy $T$, of which the cost is bounded by $\alpha$-factor of the optimal (OPT) hierarchy tree, i.e., $cost(T)\leq \alpha \cdot cost(OPT)$, to a polylogarithmic-approximate fair hierarchical clustering. This work is developed based on the ideas by Knittel et al. [2023] but simplifies some of the key operators, e.g., tree holding operator, and achieves the polylogarithmic approximation for cost for the first time.


**Strengths:**

- Proposed an algorithm that modifies an unfair hierarchical clustering and yields a fair hierarchy with only a modest increase in cost.
- Provided real dataset experiments showing that their proposed algorithm improves the fairness in clustering (i.e., balances the portion of nodes from different parties) with a modest cost ratio increase.


**Weaknesses:**

- I think the presentation of this paper needs to be largely modified. First, this paper addresses some technical terminologies before there are defined, which makes it hard to understand what the statements mean before reading until the end of the paper. For example, in Introduction the last paragraph, “O(function(N)))-approximation”is used to compare previous methodologies. But, neither the definition of the cost in hierarchical clustering nor what this approximation means was provided. Theorem 1 in page 3 is an exact same copy of the same theorem in page 5. But just presenting the same theorem without even defining what approximation means and how $\alpha_i$ and $\beta_i$ are related to the quantification of fairness makes it very hard to understand the meaning of the paper in introduction. If the authors want to summarize the key contributions in introduction, they need to properly define the terminologies and important parameters to make it self-contained.
- Another issue is related to the reference. The techniques in this paper is highly dependent on the previous techniques by Knittel et al [2023] and the authors refer the paper many times. Especially in Sec. 2.3 when tree operators are introduced, they refer the same operators introduced in Knittel. But it is hard to see what was the original way these operators were defined, what are the main changes this paper addresses, and why this change generates a better approximation.
- In experiment Fig.4, the ratio of cost increase due to fair clustering is about 8x when n is about 10^3. Can we call this a modest cost increase? Intuitively, isn't it relatively easier to achieve the fairness for larger $n$ since there are many nodes and balancing those nodes without hurting the cost can be easier due to much more ways to modify the tree? Can the authors compare their result with other algorithms in terms of achievable fairness and cost increase? There is no baseline comparison in the experiment. Also, it is not clear whether the performance of the proposed algorithm is robust to the change of the original hierarchical clustering algorithm. In the experiment, only the average-linkage was considered as an initial algorithm.



**Questions:**

- Can the authors provide experiments to compare their performance with other baselines?
- Can the authors show that the performance of proposed algorithm is robust to the change of the originally given (unfair) hierarchical clustering?


**Limitations:**

Limitations are not well addressed by the authors.

---

> ### Author Rebuttal · Authors · 2023-08-09
>
> *“I think the presentation of the paper needs to be largely modified…”*
>
> We will revise the paper for the final draft per the suggestions of all reviewers. Notably, we will guarantee that all formal theorems and definitions have a high level intuitive explanation in the introduction to alleviate the mentioned confusions.
>
> We furthermore refer the reviewer to the response of reviewer XyRG, wherein they mention that such “impenetrable” theorem statements cannot be avoided for this problem. To alleviate this, we provide some informal theorem statements and terminology definitions which will be included in the main text to improve readability (see our response to XyRG). We additionally refer to the response to reviewer qPF4’s question to better understand the results.
>
> *“What are the main changes this paper presents [as compared to Knittel 2023]...”*
>
> Our work builds upon that of Knittel 2023 in two ways. First, we reduce the number of tree operations used from 4 to 2, where the more complicated operation, tree folding, is vastly simplified. Second, in contrast to Knittel 2023 who applied one operator to the entire hierarchy at a time, we alternate applying one operator and the other. This reduces the number of top-bottom passes of the algorithm from 4 to 1 and avoids an exponential (in $O(\delta \log n)$) increase in the cost. This reason is somewhat complex and is a main technical contribution. Moreover, our analysis is considerably more readable and provides the much desired polylogarithmic cost approximation without the additional n^\delta term of the former result.
>
> *“Can we call this a modest cost increase… isn’t it easier to achieve the fairness for larger $n$…?”*
>
> Given the hardness of the vanilla problem (i.e., it’s APX-hard and the best efficient approximation is $O(\sqrt{\log n})$), and the increased complication posed by the fairness constraint, we would expect the cost to necessarily be increased by a large factor in the worst case (see the response to XyRG). We say “modest” because the cost increase is very small compared to $n$, which reflects the theory that it is at worst polylogarithmic in $n$. While larger n may in some ways give us more leeway to combine clusters, it is extremely difficult to characterize the most adversarial examples, which is an open question. Notice that our algorithms are agnostic to the degree of color representation imbalance in clusters - as in, the only time it considers colors at all is when it orders cluster by their relative representations of a fixed color. More involved, complicated algorithms will likely be needed to leverage the large cluster sizes to carefully handle these imbalances.
>
> *“Compare to other baselines…”*
>
> This is a novel problem setting and the only known algorithms for such a fairness guarantee are the present work and the cited Knittel 2023 paper. Here, we provide some experimental results contrasting the two algorithms' cost returns after producing a fair clustering for full comparison.
>
> The table below compares the relative (to unfair clustering) cost of our algorithm as compared to the baseline algorithm of Knittel 2023 (with the parameter of $\delta$ set to ¼):
>
> | n | Baseline | Our Algorithm |
> | -- | --------- | ---------------- |
> | 128 | 3.65240727 | 1.08586718 |
> | 256 | 5.68329859 | 1.42082465 |
> | 512 | 12.30571685 | 2.5869583 |
> | 1024 | 25.17125835 | 6.6745378 |
> | 2048 | 52.93911771 | 7.86944693 |
>
> As you can see, since our algorithm is only polylogarithmic in $n$, it naturally scales much better with the input sample size. This further highlights the impact of our work and will be included in the final version of our paper.
>
> *“Robust to changes in the original hierarchical clustering?”*
>
> We thank the reviewer for noting that it would be interesting to study the fair clustering problem in a dynamic setting wherein an algorithm can handle updates to the underlying structure of the clustering. This is an important problem but outside the scope of the present work. Though, with some speculation, it may be possible to add a small amount of randomness to our algorithm (e.g., whenever we compare two cluster sizes) to achieve robustness to local changes while still achieving only marginally worse approximations and degrees of fairness.

---

> > ### Author Response · Authors · 2023-08-17
> >
> > Dear reviewer,
> >
> > We hope this message finds you well. We wanted to check if you’ve had a chance to review our rebuttal for the paper. If you have any further questions or if there’s anything you’d like to discuss, please feel free to comment further. We are eager to address any concerns you might have. Thank you for your time and consideration.

---

> > > ### Comment · Reviewer_gEkD · 2023-08-18
> > > **Response to Rebuttal**
> > >
> > > Thanks for the authors' response and the extra experimental results. I think the numerical comparison between the proposed algorithm and Knittel 2023 in terms of $n$ will be a helpful addition to this paper. I also hope that the authors can incorporate the suggested changes in the next version of this paper, especially a high level intuitive explanation of the main contribution in the introduction and a careful revision for all the notations to make sure that they are properly defined in the right positions (at least before they are used in presenting the theorems). Even though I think this paper has a potential for publications in a proper venue after a careful revision, at the current state of this paper, I would keep my score.

---

### Author Response · Authors · 2023-08-16

We thank the reviewers for their insightful feedback and hope that our responses are sufficient to resolve any concerns. Please let us know if further clarifications are required.

---

### Decision · Program_Chairs · 2023-09-21

**Decision:**

Accept (poster)

**Comment:**

The paper studies the problem of fair hierarchical clustering and improves over existing work of Knittel et al.'2023 to present the first poly-logarithmic approximation to the problem. Overall the reviewers liked the paper and agreed that it is a substantial theoretical contribution. One reviewer felt that the techniques borrowed heavily from prior work of Knittel et al.' 2023. While a lot of the machinery is built on top of prior work, there is enough technical novelty in achieving the poly-log factor via the improved algorithm. The authors are encouraged to simplify the presentation more and multiple reviewers felt that in some cases the terminology was not easy to understand without looking at prior work.